# Retrospective Observational Study of CSF-Derived HIV-1 Tat and Vpr Amino Acid Sequences in a South African Pediatric Cohort with HIV Subtype C

**DOI:** 10.3390/ijms26115008

**Published:** 2025-05-22

**Authors:** Anicia Thirion, Shayne Mason, Du Toit Loots, Regan Solomons, Monray Edward Williams

**Affiliations:** 1Biomedical and Molecular Metabolism Research (BioMMet), North-West University, Potchefstroom 2520, South Africa; niciajr@gmail.com (A.T.); shayne.mason@nwu.ac.za (S.M.); dutoit.loots@nwu.ac.za (D.T.L.); 2Department of Paediatrics and Child Health, Faculty of Medicine and Health Sciences, Stellenbosch University, Tygerberg 7505, South Africa; regan@sun.ac.za

**Keywords:** HIV-1, Tat, Vpr, sequencing, pediatric

## Abstract

The human immunodeficiency virus (HIV-1) infiltrates the central nervous system (CNS) early in infection, leading to HIV-associated neurocognitive impairments, particularly pronounced in children who exhibit neurodevelopmental delay. Viral proteins, including the transactivator of transcription protein (Tat) and viral protein R (Vpr) are pivotal in HIV-1 neuropathogenesis, with their amino acid sequence variation influencing disease progression. Due to the difficulty of collecting cerebrospinal fluid from children, few studies have examined whether key Tat and Vpr neuropathogenic signatures found in blood are also present in the cerebrospinal fluid (CSF) of children with HIV. We employed Sanger sequencing for Tat and Vpr sequence analysis using retrospectively collected CSF samples from a South African pediatric HIV-1 subtype C cohort (n = 4). We compared our CSF-derived sequences with pediatric blood-derived sequences (n = 43) from various geographical regions, sourced from the Los Alamos database. Neuropathogenic amino acid variants were identified in Tat and Vpr sequences derived from CSF samples of South African pediatric participants No significant differences were found between subtype C sequences from CSF and blood. Regional analysis highlighted unique amino acid signatures. Obtaining pediatric CSF for HIV-1 sequencing is highly challenging. Despite a small sample size, this study offers rare insights into Tat and Vpr sequences in children, improving understanding of the potential HIV-1 brain pathogenesis in pediatric populations.

## 1. Introduction

The human immunodeficiency virus 1 (HIV-1) is the dominant strain globally [1], with around 1.5 million children infected, as reported by UNAIDS in 2023 [2]. HIV-1 can enter the central nervous system (CNS) early in infection via infected immune cells crossing the blood-brain barrier (BBB) [3]. While combination antiretroviral therapy (cART) lowers plasma viral load, it may not fully penetrate the BBB, allowing continued viral replication in the CNS, a phenomenon known as cerebrospinal fluid (CSF) viral escape [3]. Studies show that HIV-1 induces chronic neuroimmune activation and neurometabolic dysregulation [4,5]. Untreated children display higher rates of CNS dysfunction than untreated adults [6], likely due to the impact of the virus on brain development. Neurodevelopmental delays and cognitive impairments in children with HIV vary in severity, influenced by HIV-1 subtype differences [7].

HIV-1 subtype variations present with altered viral protein amino acid sequences, leading to differences in neurovirulence and neurological outcomes [7]. The HIV transactivator of transcription protein (Tat) and viral protein R (Vpr) cooperate to regulate viral genome transcription and cell apoptosis [8]. Tat, secreted early in replication, persists even under cART and contributes to neurotoxicity [9,10]. Several subtype-specific amino acid mutations in Tat, including mutations from lysine to asparagine at residue 24 (K24N), arginine to histidine at residue 29 (R29H), cysteine to serine at residue 31 (C31S), and arginine to serine at residue 57 (R57S), have been associated with neuropathogenesis [11,12,13]. Recent studies by our group in South African adults with HIV highlighted the relative frequencies of these substitutions. At position 24, lysine (K) was observed in 37% of sequences, asparagine (N) in 28%, and proline (P) in 9% [14]. At position 31, serine (S) appeared in 66% of sequences and cysteine (C) in 11%. At position 57, serine (S) was found in 53%, arginine (R) in 9%, and alanine (A) in 8% of sequences [14,15,16,17,18].

Vpr contributes to HIV-1 neuropathogenesis by altering CNS metabolism and communication [19]. Vpr functions include virion incorporation, oligomerization, nuclear transport, differentiation, cell cycle arrest, and apoptosis [9,19]. Subtype-specific amino acid mutations in Vpr, including mutations from glutamate to isoleucine at residue 21 (E21I), glycine to serine at residue 41 (G41S), serine to asparagine at residue 41 (S41N), tyrosine to histidine at residue 45 (Y45H), alanine to threonine at residue 55 (A55T), and arginine to glutamine at residue 77 (R77Q), have been associated with neuropathological effects [16,17,20,21,22]. Recent studies by our group in South African adults with HIV highlighted the relative frequencies of these substitutions. At position 45, tyrosine (Y) was present in 53% of sequences, while histidine (H) appeared in 33%. At position 55 of Vpr, threonine (T) was observed in 70% and alanine (A) in 19% of sequences [14]. At position 77, glutamine (Q) was present in 84%, arginine (R) in 6%, and histidine (H) in 5% of sequences [14,15,16,17,18].

HIV replication and viral load in the CNS are typically lower than in systemic circulation. While some studies have determined Tat and/or Vpr amino acid sequences in the CNS from adults [23,24], few have focused on pediatric populations due to the challenges associated with CSF collection. As a result, little is known about whether the predominant CNS-derived virus in pediatric HIV exhibits similar Tat and Vpr amino acid sequence variations to those found in pediatric blood-derived virus. Given the rapid disease progression and severe neurocognitive impairments in children with HIV, understanding Tat and Vpr genetic diversity is crucial within this compartment. Therefore, this study aimed to (1) analyze CSF-derived Tat and Vpr sequences in a pediatric HIV-1C cohort and (2) compare them with pediatric blood-derived sequences from different regions.

## 2. Results

### 2.1. Study Characteristics

CSF viral sequences were obtained from n = 4 participants (C-SA_s). Similar sample sizes have been reported in studies sequencing HIV-1 in adult CSF [25,26] and pediatric plasma [27]. Ethical challenges in pediatric populations often result in smaller cohorts. The mean age of participants was 16.25 months (±15.26), consisting of two males and two females, with a mean CD4+ count of 680.75 cells/µL (±597.27). Viral load data were available only for one participant, who had 49,000 copies/mL; three participants were untreated, while one was on ART (Appendix A). Briefly, Participant C-SA_1 was an 8-month-old who presented with miliary tuberculosis, pneumonia, gastroenteritis, signs of meningeal irritation, and neurological symptoms including decreased consciousness and seizures. Participant C-SA_1 was ART-naïve, with low CD4 counts and a detectable viral load, suggestive of severe immune suppression and active CNS infection (Appendix A). Participant C-SA_2, an 11-month-old, exhibited signs of peripheral tuberculosis (TB) and meningeal irritation. Although viral load data were unavailable, Participant C-SA_2 had a high CD4 count, suggesting relatively preserved immune function. Participant C-SA_3, a 39-month-old, presented with co-infections including miliary TB and lymphocytic interstitial pneumonitis, along with an elevated viral load, indicating a complex disease presentation. Lastly, Participant C-SA_4 was a 7-month-old who was receiving ART at the time of evaluation and had high viral load, HIV encephalopathy, and disseminated TB (Appendix A).

### 2.2. Subtype C Tat and Vpr Sequences in Study Participants

HIV-1 subtyping indicated that all participants in this cohort (C-SA_s) belonged to subtype C. The nucleotide sequences of Tat exon 1 (216 bp) and Vpr (288 bp) were translated into amino acid sequences of 72 and 96 residues, respectively (Appendix A). These sequences have been uploaded to GenBank under accession numbers PP437871–PP437877.

Several conserved regions were identified in the Tat amino acid sequences. Positions 24, 29, 31, and 57 have previously linked to Tat neuropathogenesis. In our cohort (Tat-C-SA_s), substitutions associated with increased neuropathogenesis were observed at K24 (2/4; 50%), R29 (1/4; 25%), and R57 (1/4; 25%). Conversely, substitutions associated with reduced neuropathogenesis included H29 (1/4; 25%), S31 (4/4; 100%), and S57 (3/4; 75%). Ambiguous amino acids were identified at positions 2, 58, 59, 61, 63, and 69, suggesting underlying viral population diversity; however, key Tat amino acids were still clearly defined.

Several conserved regions were identified in the Vpr amino acid sequences. Positions 21, 41, 45, 55, and 77 have previously been linked to Vpr neuropathogenesis. In our cohort (Vpr-C-SA_s), substitutions associated with increased neuropathogenesis were observed at S41 (1/3; 33%) and Y45 (2/3; 66%). Conversely, substitutions associated with reduced neuropathogenesis included E21 (2/3; 66%), G41 (1/3; 33%), H45 (1/3; 33%), T55 (2/3; 66%), and Q77 (2/3; 66%). Ambiguous amino acids were identified at positions 22, 33, 35, 37, 46, 57, 61, 64, 68, 85, 89, and 91, however, key Vpr amino acids were still clearly defined.

Given the high replication rate and error-prone nature of HIV the reverse transcriptase, HIV typically exists as a quasispecies within individuals [12]. During inspection of the Sanger sequencing chromatograms, we observed double peaks at certain nucleotide positions, consistent with the presence of mixed viral populations. For the Tat fragment, the nucleotide ambiguity rate was less than 2.5%, and for the Vpr fragment, it was less than 5.59%.

To ensure accuracy, chromatogram peak heights were manually reviewed: minor secondary peaks (likely background noise) were resolved through manual base calling, while sites with equally prominent peaks were retained as ambiguous to reflect true viral diversity. As is standard in HIV Sanger sequencing studies, the resulting sequences represent consensus genomes derived from mixed viral populations. While the resulting sequences represent consensus genomes derived from mixed viral populations, it is important to note that at the key neuropathogenic amino acid positions of interest, there was a clear consensus, with no ambiguities observed. Therefore, despite the background sequence variability, we have high confidence in the accuracy of the critical residues analyzed in this study.

### 2.3. Comparison Between South African Subtype C CSF-Derived and PBMC-Derived Tat and Vpr Sequences

Since only one pediatric CSF-derived sequence was available in the Los Alamos database, no meaningful comparisons within this compartment were possible. Therefore, we compared our CSF-derived sequences to blood-derived sequences. We evaluated sequence similarity between CSF-derived sequences from our cohort (Tat/Vpr-C-SA_s) and peripheral blood mononuclear cells (PBMCs)-derived sequences from another pediatric South African cohort (Tat/Vpr-C-SA_EU) in the Los Alamos database. Phylogenetic analysis (Figure 1A) showed that while Tat-C-SA_3 was less related to other South African sequences, all remaining sequences, whether from PBMCs or CSF, were closely related.

Alignment of all Tat amino acid sequences (Appendix A) revealed unique variants at various frequencies in PBMC-derived sequences that were not present in our cohort, including P3L (1/5; 20%), N7K (1/5; 20%), N12K (1/5; 20%), L35Q (2/5; 40%), Q39L (2/5; 40%), and P59S (1/5; 20%). Analysis of neuropathogenic Tat signatures in PBMC-derived sequences showed K24 (3/5; 60%), R29 (1/5; 20%), C31 (2/5; 40%), and no R57 variants. A chi-square test corrected using the false discovery rate (FDR) found no statistically significant differences in the frequency of Tat neuropathogenic signatures between CSF-derived sequences from our cohort and blood-derived sequences from the Los Alamos database (Appendix A), suggesting no significant differences between the groups.

Inspection of the phylogenetic tree of the Vpr amino acid sequences (Figure 1B), revealed that participant Vpr-C-SA_3 was less closely related to all other South African sequences which clustered together regardless of whether they were derived from CSF or PBMCs. Alignment of all Vpr amino acid sequences (Appendix A) revealed unique variants at various frequencies in the PBMC-derived sequences not present in our cohort sequences, including A4P (5/5; 100%), Y15H (1/5; 20%), A19T (2/5; 40%), L22V (3/5; 60%), E25D (1/5; 20%), L42I (1/5; 20%), E58Q (1/5; 20%), I61T (1/5; 20%), I74L (1/5; 20%), and S94G (1/5; 20%). Residues A59 and L84 were conserved in PBMC-derived but variable in CSF-derived Vpr. Neuropathogenic Vpr signatures in PBMC-derived sequences showed S41 (4/5; 80%), Y45 (4/5; 80%), and A55 (2/5; 40%) but no I21, N41, or R77 substitutions. A chi-square test identified significant differences between specific CSF- and blood-derived Vpr sequences (Appendix A); however, after FDR correction for multiple comparisons, no significant differences were observed. Despite the limited scope of our investigation, these findings suggest similarities between CNS and peripheral HIV-1 compartments for Tat and Vpr in children.

### 2.4. Comparison Between CSF-Derived Subtype C Sequences and PBMC-Derived Tat and Vpr Sequences Across Various Geographical Regions

We assessed whether CSF-derived subtype C Tat/Vpr sequences from our cohort differed from PBMC-derived sequences from pediatric cohorts in various geographical regions. To evaluate Tat sequence similarity, we compared our CSF-derived subtype C Tat sequences (Tat-C-SA_s; n = 4) to PBMC-derived sequences from South Africa (Tat-C-SA_EU; n = 5), India (Tat-C-IN_KF; n = 27), and the USA (Tat-B-USA_MK, M, U, AYB; n = 10) in the Los Alamos database. The phylogenetic tree (Figure 2A) revealed distinct clustering for USA subtype B sequences, as expected. Indian subtype C sequences also clustered distinctly, while South African sequences, including our cohort, exhibited a broader distribution. Another maximum likelihood phylogenetic tree (Appendix A) was constructed using representative sequences from each region (randomly selected, as done in previous studies [28,29]) but no separation was observed between Indian and South African sequences.

Comparing pediatric Tat amino acid sequences across geographical regions revealed significant differences after correction using FDR (Appendix A). Neurotoxic K24 was more frequent in South African Tat compared to Indian and USA sequences (*p* < 0.001), while C31 and R57 were more prevalent in USA Tat (*p* < 0.001). Subtype C (South African and Indian) sequences had significantly higher frequency of E2 and D64, whereas USA sequences had a higher frequency of K12, T23, F32, D61, Q63, and L69 (*p* < 0.001 for all). In contrast, N12, N23, Y32, S61, E63, and I69 were more common in South African and Indian sequences (*p* < 0.001 for all).

We compared CSF-derived subtype C Vpr sequences from our cohort (Vpr-C-SA_s) with pediatric PBMC-derived Vpr sequences found in the Los Alamos database from South Africa (Vpr-C-SA_EU), the USA (Vpr-B-USA_MK, M, U, AYB), and Portugal (Vpr-B-PR_KM) using a maximum likelihood phylogenetic tree with representative variants included (Figure 2B). Examination of the unrooted phylogenetic tree revealed that South African sequences were closely related, while USA sequences were more distantly related. As expected, the Portugal sequence clustered with other subtype B regions. Among South African sequences, Vpr-C-SA_4 was less closely related, likely due to ART exposure.

Comparison of Vpr amino acid sequences across geographical regions revealed significant differences after correction using FDR (Appendix A). Specifically, E48 was more prevalent in USA Vpr sequences (*p* = 0.029). N28 was more abundant in USA sequences, whereas Q28 was more prevalent in South African sequences (*p* < 0.001).

## 3. Discussion

This study provides key insights into HIV-1C in South African pediatric cases. The lack of data on HIV Tat and Vpr amino acid sequence variation in the CNS compartment of children represents an important gap in current knowledge. Since pediatric HIV is usually acquired perinatally and children’s immune systems are still developing, this may influence patterns of HIV replication and evolution, allowing more time for CNS compartmentalization compared to adult-acquired HIV.

Limited studies have investigated CSF-derived sequences in pediatric HIV. A study by Sturdevant et al. (2012) [30], investigated the compartmentalization of the HIV-1 C env protein in a large cohort of Malawian children 3 years or younger and found significant genetic compartmentalization between the blood and CSF populations in 28% of subjects, with compartmentalization being significantly related to older age and a higher CSF/blood viral load ratio. We report CSF-derived subtype C Tat/Vpr amino acid sequences in a South-African population and identify several variants previously linked to neuropathogenesis. No significant differences were found between Tat/Vpr sequences from CSF and blood in South African subtype C. However, the size and age difference of the cohorts compared here limits the conclusions that can be drawn. Additionally, specific amino acid variations were associated with different geographical regions. This study addresses an important gap in research that has largely focused on adult blood and brain tissues/CSF sequences [24], with limited pediatric blood-derived sequences [31], and no prior investigations of pediatric CSF-derived sequences. Although our cohort is small, similar studies in adult CSF have had comparable sample sizes [25,32]. The scarcity of pediatric CSF studies is due to ethical concerns and the invasive nature of CSF collection [33]. A search of the Los Alamos database (1984–2024) identified only 10 studies with pediatric CSF or blood sequences, underscoring the significance of these findings. Understanding the structure-function relationships of key CNS proteins may help identify new therapeutic targets.

The viral protein Tat, associated with neurotoxicity, exhibited key amino acid variants at positions 24, 29, 31, and 57 implicated in neuropathogenesis. Variants K24 and R29 identified in this cohort have been linked to neurocognitive impairment in adults, with K24 associated with higher viral load and lower CD4+ T-cell counts [15]. Position 29 exhibited variability, with one participant showing the neurotoxic R29 variant while others had less neurotoxic histidine and alanine substitutions [15].

The Tat C31S mutation, predominant in subtype C and this cohort, was previously shown to abolish the chemotactic ability of Tat and reduce Tat-induced NMDAR activation [10,11]. While cell culture studies suggest C31S decreases neurotoxicity [34], clinical studies report similar outcomes for both C31 and S31 variants [35]. Similarly, in a pediatric Malawian cohort, C31 diversity did not significantly affect neurocognitive outcomes [31]. The role of the C31S mutation requires further investigation.

The R57 amino acid variant, common in Tat subtype B, is linked to increased neuropathogenesis, whereas the R57S substitution, more prevalent in subtype C, is associated with reduced neurotoxicity [12]. R57S correlates with lower peripheral levels of thymidine phosphorylase (TYMP) and chemokine ligand 2 (CCL2), mediators of HIV-1 (neuro)pathogenesis [36]. Thus, the predominant R57S mutation in our cohort indicates reduced neuropathogenesis.

Comparisons between South African subtype C CSF-derived and PBMC-derived Tat revealed no significant differences. Although selective pressures between blood and CNS may yield slight variations [37], studies on drug-resistant mutations also report minimal differences between these compartments [38]. There is a great need for comprehensive analyses, including whole-genome sequencing.

Comparison of CSF-derived subtype C Tat sequences with PBMC-derived sequences from different regions showed that the neurotoxic variant K24 was more common in South African Tat than in Indian or USA sequences. In contrast, USA sequences had higher frequencies of neurotoxic variants C31 and R57. These results align with studies suggesting that subtype B Tat is more neurotoxic than subtype C, leading to greater neuronal damage [7]. Significant differences between subtype C and B Tat sequences were found at positions 2, 12, 23, 32, 61, 63, 64, and 69, which may affect HIV neuropathogenesis. For example, the E2D substitution reduces cellular uptake and neurotoxicity [39], while residues 12, 23, and 61 are involved in the transactivation of Tat [11]. Variants at positions 63, 64, and 69 may influence the apoptotic activity of Tat and merit further investigation [40].

Vpr amino acid mutations at positions 21, 24, 41, 45, 55, and 77 have been linked to neuropathology. Notably, mutations G41S, H45Y, and T55A have been associated with altered CNS metabolism and neurotoxic quinolinic acid accumulation [16,17]. In our cohort, G41 and S41 variants occurred at similar frequencies, with Y45 appearing twice as often as H45, and T55 variants present in all participants. The Q77 variant, associated with long-term non-progression and HIV-associated neurocognitive disorders (HAND), was also identified [20,22].

Our findings suggest that Vpr exhibits similar characteristics in the CNS and periphery, though confirmation through whole-genome sequencing is necessary. A comparison of CSF-derived subtype C Vpr sequences with PBMC-derived sequences from South Africa, Portugal, and the USA indicated that only residue 45 exhibited significant differences, with the neurotoxic variant Y45 being more prevalent in South African sequences. This supports previous research indicating that subtype B Vpr is more neurotoxic than subtype C [7].

### Limitations of the Study

This study has several limitations. First, the small sample size reflects the difficulty of obtaining pediatric CSF samples due to ethical concerns and low CNS viral load, making large cohorts challenging to assemble. Our goal was to provide a snapshot of Tat/Vpr sequence variation in a rare pediatric CSF cohort, consistent with prior small-scale studies. Considering the high heterogeneity of HIV-1 viral sequences, our findings should be interpreted with this in mind. Second, our participants were younger than those in the available database sequences. Due to limited availability of pediatric sequences, we included samples across all age groups for comparative purposes. Third, differences in cART exposure may have introduced varying evolutionary pressures. Additionally, database sequences were collected over different periods (1984–2014) compared to ours (2010–2017). We also compared our CSF-derived sequences to blood-derived sequences from other participants using data from the Los Alamos HIV Sequence Database. While this comparison provided some context, a direct comparison between our CSF-derived sequences and blood-derived sequences from the same patients would have been more informative. However, due to limited sample availability, this was not possible. Future studies should consider investigating matched CNS and PBMC samples to further elucidate the dynamics of HIV compartmentalization and its impact on disease progression. This study highlights two key issues: (1) HIV-1 sequencing is often deprioritized due to assumptions of subtype or regional uniformity, and (2) sequences are not consistently deposited in repositories. Lastly, our analysis reflects the viral population diversity within the sample; however, Sanger sequencing has inherent limitations in detecting and interpreting quasispecies. The high mutation rate of HIV inevitably leads to sequence ambiguities, and despite manual chromatogram review, some errors may persist, particularly at low-quality positions. The limited depth of Sanger sequencing makes it challenging to distinguish low-frequency variants from sequencing artifacts. Although we retained ambiguous base calls where peak intensities were similar, this approach may not fully capture the viral complexity. For finer resolution of rare variants, next-generation sequencing would be more suitable. Nevertheless, for the neuropathogenic amino acid positions analyzed, no ambiguities were present, and we are therefore confident in the data presented for these specific signatures. Despite these challenges, this study offers valuable insights into HIV genetic diversity in the developing brain and across regions.

## 4. Materials and Methods

### 4.1. Sample Characteristics

CSF samples were retrospectively collected for routine diagnostics from children suspected of meningitis at Tygerberg Hospital, Cape Town, South Africa, between 2010 and 2017. Lumbar puncture was performed on 12 HIV-positive (HIV+) pediatric cases, with HIV screening conducted as previously described [4,5]. Written informed consent/assent was obtained from participants or their guardians for the inclusion of clinical and investigative data. The study was approved by the Health Research Ethics Committee (HREC) of Stellenbosch University, Tygerberg Hospital (ethics approval no. N16/11/142), the Western Cape Provincial Department of Health and Wellness, and the HREC of North-West University (NWU), Potchefstroom campus (ethics approval no. NWU-00,063-18-A1). The CSF samples were frozen and transported overnight to the Centre for Human Metabolomics at NWU, where they were stored in a dedicated −80 °C freezer in a Biosafety Level 3 laboratory until analysis. Samples were filtered using Amicon Ultra-2 mL 10,000 MWCO centrifugal filters at 4500× *g* for 20 min. Each pelleted sample was resuspended in 400 µL phosphate-buffered saline, with 200 µL aliquoted for RNA extraction and sequencing.

### 4.2. Laboratory Assessment of CSF

RNA was extracted using the Quick-RNA™ viral kit (Zymo Research, Irvine, CA, USA). This kit offers a streamlined process that effectively lyses viral particles and inactivates nucleases and infectious agents, ensuring the integrity of the RNA. The specialized buffer system facilitates complete viral particle lysis, enabling efficient RNA isolation from samples containing various viruses, including HIV. The isolated high-quality viral RNA is ready for downstream applications. RNA was reverse transcribed with the ProtoScript^®^ II First Strand cDNA Synthesis Kit (New England Biolabs, Ipswich, MA, USA). DNA was prepared for PCR amplification of Tat exon 1/Vpr/Vif (HXB2 positions 4900–6351) using primers Vif-1 (5′ GGGTTTATTACAGGGACAGCAGAG) and CATH-4R (5′ GTACCCCATAATAGACTGTGACC) according to previously described protocols [28]. PCR products were purified with the NucleoSpin^®^ Gel and PCR Clean-up Kit (Machery-Nagel GmbH & Co.KG, Düren, Germany). Sequencing was performed using the BigDye Terminator v.3.1 Cycle Sequencing Ready Reaction Kit (ThermoFisher Scientific, Waltham, MA, USA) and analyzed on an ABI Prism 3130xl DNA sequencer (Applied Biosystems, Foster City, CA, USA). Of the 12 children with HIV, sequencing was successful for only 4 participants (33%), designated as Tat-C-SA_ID or Vpr-C-SA_ID in this study. This limited success was likely due to one or more of the following factors: (1) RNA degradation between the time of sample collection (2010–2017) and analysis (2021); (2) low viral load in the cerebrospinal fluid (CSF); and/or (3) low amplification efficiency, possibly related to suboptimal RNA quality or mismatches in primer binding regions.

### 4.3. Bioinformatics Analysis

Sequences were analyzed using GeneStudio™ professional sequence analysis software (Version 2.2). Nucleotide sequences were translated into amino acid sequences with ExPASy translate. The study cohort was subtyped using COMET [41]. Consensus amino acid sequence alignments were then constructed with CLC Genomics Workbench 24.0 (https://www.qiagen.com/us/products/discovery-and-translational-research/next-generation-sequencing/informatics-and-data/analysis-and-visualization/clc-genomics-workbench (accessed on 1 March 2024)).

### 4.4. Database Sequences

HIV-1C Tat and Vpr sequences were retrieved from the Los Alamos database in March 2024 (https://www.hiv.lanl.gov/components/sequence/HIV/search/search.html (accessed on 18 May 2025)). A GenBank search with terms like “HIV”, “pediatric”, “perinatal”, “infant”, and “child”, along with a literature review, identified additional sequences. Nucleotide sequences for each group were translated into amino acid sequences using ExPASy translate. Consensus alignments were constructed using CLC Genomics Workbench 24.0. The sequences were designated as follows: Tat/Vpr-C-SA_EU (South Africa, 6 children, collected in 1999), Tat-C-IN_KF (India, 27 children, collected 2007–2011), Tat/Vpr-B-M, U, AF, and AYB (United States, 16 children, collected 1984–2013), and Vpr-B-PR_KM (Portugal, 1 child, collected in 2014).

### 4.5. Phylogenetic Analysis

Phylogenetic trees were constructed to analyze the relationships between Tat/Vpr amino acid sequences from pediatric HIV-1 cases across various geographical regions and subtypes, including this study cohort (Tat/Vpr-C-SA_s) and downloaded sequences (Tat/Vpr-C-SA_EU; Tat-C-IN_KF; Tat/Vpr-B-M, U, AF, AYB; Vpr-B-PR_KM). Analyses were conducted using CLC Genomics Workbench 24.0. Multiple sequence alignment and selection of the optimal substitution model for unrooted maximum likelihood trees were performed as previously described [28]. The identified model was applied to create an unrooted maximum likelihood tree using the “Maximum Likelihood Phylogeny” tool with 100 bootstrap replicates, visualized in “cladogram” mode to group sequences by relatedness.

### 4.6. Statistical Analysis

Statistical analyses were conducted using SPSS (version 29, IBM, New York, NY, USA). The chi-square test, corrected for multiple testing using FDR, was utilized to compare the groups and a significance level of *p* < 0.05 was deemed as statistically significant.

## 5. Conclusions

This study is the first to report CSF-derived Tat/Vpr amino acid sequences in South African children. It provides valuable insights into the frequency of specific Tat/Vpr amino acids within this context. Our findings indicate limited variation between blood and CNS compartments, while highlighting significant differences in amino acid diversity across geographical regions. Further research is needed to evaluate the impact of the identified amino acid substitutions on neurodevelopment and neurocognitive outcomes in children living with HIV.

## Figures and Tables

**Figure 1 ijms-26-05008-f001:**
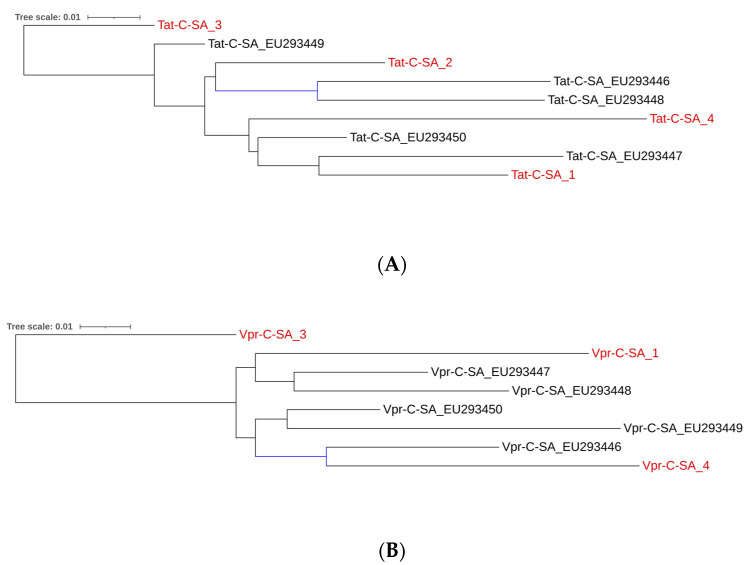
(**A**): Maximum likelihood phylogenetic tree analysis comparing Tat CSF-derived subtype sequences (red) with PBMC-derived sequences originating from South Africa constructed using the general time reversible model with gamma rate variation and topology variation (GTR + G + T). The bootstrap probability (>70%, 100 replicates) is highlighted in blue. (**B**): Maximum likelihood phylogenetic tree analysis comparing Vpr CSF-derived subtype sequences (red) with PBMC-derived sequences originating from South Africa constructed using the general time reversible model with gamma rate variation and topology variation (GTR + G + T). The bootstrap probability (>70%, 100 replicates) is highlighted in blue.

**Figure 2 ijms-26-05008-f002:**
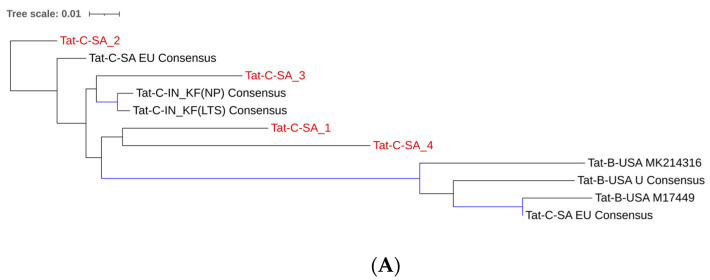
(**A**): Maximum likelihood phylogenetic tree analysis comparing consensus Tat CSF derived sequences (red) with PBMC-derived sequences originating from South Africa, India, and USA constructed using the Kimura 80 model with variation in the gamma rate and topology (K80 + G + T). The bootstrap probability (>70%, 100 replicates) is highlighted in blue. (**B**): Maximum likelihood phylogenetic tree analysis comparing consensus Vpr (CSF derived sequences (red) with PBMC-derived sequences originating from South Africa, India, and USA constructed using the Kimura 80 model with variation in the gamma rate and topology (K80 + G + T). The bootstrap probability (>70%, 100 replicates) is highlighted in blue.

## Data Availability

The data that support the findings of this study are available in the Appendix A of this article. The sequences generated as part of this study were uploaded to GenBank under accession numbers PP437871-PP437877.

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
