# Peer review of "Retrospective Observational Study of CSF-Derived HIV-1 Tat and Vpr Amino Acid Sequences in a South African Pediatric Cohort with HIV Subtype C"

_ijms, 2025, doi:10.3390/ijms26115008_

Round 1

Reviewer 1 Report

Comments and Suggestions for Authors

Thirion et al assess sequence diversity of tat and vpr genes isolated from the CSF of 4 pediatric patients. The authors state that there are very few reports that compare virus diversity between CSF and blood reservoirs in children living with HIV. Comparisons are made between the sequences derived from the 4 pediatric patients and sequences available in the LANL sequence database, with no major differences being observed between Subtype C sequences from CSF and blood. While these observations are potentially interesting, there are several issues with the conclusions drawn and the rigor of the analysis. First, there are no direct comparisons made between the patient CSF samples characterized herein and matched virus isolated from peripheral blood of these patients, which was the deficiency in the field outlined by the authors. Second, a majority of the sequences isolated contain several stop codons, which suggests these were derived from replication-incompetent virus. This skews the functional correlation/selection of these isolates in vivo as the viruses would be selected against. Thus, it is unclear how to correctly interpret the presence of these sequences in comparison to those listed in LANL. Third, while it is fully appreciated that getting CSF from pediatric patients is extremely difficult, having such a low n-value for a virus as heterogeneous as HIV severely limits any conclusions that can be drawn from the sequences, especially given that a majority contain stop codons.

Major:

  • While it is understandable that getting CSF samples from children is difficult, assessing diversity in only 4 patients severely limits possible conclusions that can be drawn given the heterogenous nature of HIV within patients and across patient populations

  • How was the virus actually isolated from the CSF? I couldn’t find those details in the manuscript, and they are particularly important given that these sequences seem to originate from replication-defective virus.

  • How many clones were sequenced per patient? Only 4 Tat and 3 Vpr sequences are reported. Are these consensus sequences derived from multiple clonal fragments isolated from the CSF? The PCR fragments should have bene clonally isolated and sequenced individually to determine nucleotide heterogeneity at each position.

  • No matched comparison between virus isolated from the CNS and PBMCs from the same patients. This is particularly important because of the point below.

  • It is not clearly defined that “X” in the protein coding sequences provided in the supplemental indicates a stop codon. Three of the four Tat open reading frames and all three of the Vpr open reading frames have stop codons, and several sequences have 2+ stop codons. This likely indicates that these sequences were isolated from replication defective virus (especially the stop codon at position 2 in Tat isolate SA_2). These observations underscore the need for more rigor in this study to assess the true virus diversity in this “cohort.”
  • Given that 6/7 of the isolates here have stop codons, and thus are by default selected against, what can be gained from the phylogenetic comparisons with sequences in the LANL database given that those isolates do not have stop codons? In other words, polymorphisms in these deleterious proteins may not be functionally relevant as it could be a dead virus.

Author Response

 13 May 2025

We would like to extend our sincere thanks to the reviewers and the editorial board for their careful evaluation of our manuscript. We are grateful for the constructive comments and valuable suggestions, which have helped us to improve the clarity, focus, and overall quality of the work. We appreciate the opportunity to revise our manuscript in response to the feedback, and we have carefully addressed each point below. The revisions made in response to their feedback are detailed below and have been appropriately highlighted throughout the manuscript.

Reviewer 1

Thirion et al assess sequence diversity of tat and vpr genes isolated from the CSF of 4 pediatric patients. The authors state that there are very few reports that compare virus diversity between CSF and blood reservoirs in children living with HIV. Comparisons are made between the sequences derived from the 4 pediatric patients and sequences available in the LANL sequence database, with no major differences being observed between Subtype C sequences from CSF and blood. While these observations are potentially interesting, there are several issues with the conclusions drawn and the rigor of the analysis. First, there are no direct comparisons made between the patient CSF samples characterized herein and matched virus isolated from peripheral blood of these patients, which was the deficiency in the field outlined by the authors. Second, a majority of the sequences isolated contain several stop codons, which suggests these were derived from replication-incompetent virus. This skews the functional correlation/selection of these isolates in vivo as the viruses would be selected against. Thus, it is unclear how to correctly interpret the presence of these sequences in comparison to those listed in LANL. Third, while it is fully appreciated that getting CSF from pediatric patients is extremely difficult, having such a low n-value for a virus as heterogeneous as HIV severely limits any conclusions that can be drawn from the sequences, especially given that a majority contain stop codons.

  1. While it is understandable that getting CSF samples from children is difficult, assessing diversity in only 4 patients severely limits possible conclusions that can be drawn given the heterogenous nature of HIV within patients and across patient populations.

Response: We fully acknowledge this limitation and have made a substantial effort to transparently address it throughout the manuscript. We concur with the reviewer that the small sample size constrains the generalizability of our findings. However, we believe our study contributes valuable insights into the under-researched area of pediatric HIV-1 sequences, particularly within the South African context.

To emphasize this point, we have explicitly acknowledged or eluded to the sample size limitation in several sections of the manuscript:

Line 28-29: “Despite a small sample size, this study offers rare insights into Tat and Vpr sequences in children, improving understanding of the potential HIV-1 brain pathogenesis in pediatric populations.”

Line 252-254: “Although our cohort is small, similar studies in adult CSF have had comparable sample sizes [25, 31]. The scarcity of pediatric CSF studies is due to ethical concerns and the invasive nature of CSF collection [32].”

Line 314 – 319: “First, the small sample size reflects the difficulty of obtaining pediatric CSF samples due to ethical concerns and low CNS viral load, making large cohorts challenging to assemble. Our goal was to provide a snapshot of Tat/Vpr sequence variation in a rare pediatric CSF cohort, consistent with prior small-scale studies. Considering the high heterogeneity of HIV-1 viral sequences, our findings should be interpreted with this in mind”

  1. How was the virus actually isolated from the CSF? I couldn’t find those details in the manuscript, and they are particularly important given that these sequences seem to originate from replication-defective virus.

Response: We appreciate the opportunity to clarify our methodology. In Section 6.1 (line 356) of our manuscript, we describe the process from CSF samples collect to removing large debris molecules to prepare a sample suitable for viral RNA extraction. To achieve viral RNA extraction, we utilized the Quick-RNA™ Viral Kit (Zymo Research), which provides a streamlined process for isolating high-quality viral RNA from cerebrospinal fluid (CSF) samples. The Quick-RNA™ Viral Kit employs a specialized buffer system that facilitates complete viral particle lysis, effectively inactivating nucleases and infectious agents. This ensures the integrity of the RNA during extraction. The kit is optimized for low viral copy detection, making it suitable for sensitive downstream applications such sequencing and PCR. Additionally, the inclusion of DNA/RNA Shield™ in the kit ensures nucleic acid stability during sample storage and transport at ambient temperatures (4–25°C).

We have added the following:

Line 374-379 “This kit offers a streamlined process that effectively lyses viral particles and inactivates nucleases and infectious agents, ensuring the integrity of the RNA. The specialized buffer system facilitates complete viral particle lysis, enabling efficient RNA isolation from samples containing various viruses, including HIV. The isolated high-quality viral RNA is ready for downstream applications.”

  1. How many clones were sequenced per patient? Only 4 Tat and 3 Vpr sequences are reported. Are these consensus sequences derived from multiple clonal fragments isolated from the CSF? The PCR fragments should have bene clonally isolated and sequenced individually to determine nucleotide heterogeneity at each position.

Response:  We appreciate the reviewer's suggestion regarding clonal sequencing. Due to the limited volume of CSF samples and the inherent difficulty in isolating sufficient virus for clonal analysis, we employed a bulk PCR approach to amplify the Tat and Vpr genes from the CSF samples, followed by direct Sanger sequencing of the PCR amplicons. This approach allowed us to obtain sequences representing a mixture of viral variants present in each sample. While clonal sequencing would have provided a higher-resolution assessment of nucleotide heterogeneity, our study aimed to determine whether the predominant HIV-1 variants present in blood-derived samples were also present in pediatric CSF samples. Despite the inherent viral diversity in HIV sequences, the amino acid positions of interest, associated with neuropathogenic signatures, were predominant in all sequences analyzed. We acknowledge the limitations of our approach and have transparently addressed them in the manuscript.

First we acknowledge and provided context in the results section

Line 115-130 “Given the high replication rate and error-prone nature of HIV the reverse transcriptase, HIV typically exists as a quasispecies within individuals [12]. During inspection of the Sanger sequencing chromatograms, we observed double peaks at certain nucleotide positions, consistent with the presence of mixed viral populations. The Tat fragment, the nucleotide ambiguity rate was less than 2.5%, and for the Vpr fragment, it was less than 5.59%.

To ensure accuracy, chromatogram peak heights were manually reviewed: minor secondary peaks (likely background noise) were resolved through manual base calling, while sites with equally prominent peaks were retained as ambiguous to reflect true viral diversity. As is standard in HIV Sanger sequencing studies, the resulting sequences represent consensus genomes derived from mixed viral populations. While the resulting sequences represent consensus genomes derived from mixed viral populations, it is important to note that at the key neuropathogenic amino acid positions of interest, there was a clear consensus, with no ambiguities observed. Therefore, despite the background sequence variability, we have high confidence in the accuracy of the critical residues analyzed in this study.”

We acknowledge that our study design did not permit an in-depth evaluation of the heterogeneity of sequences within this cohort. Such an analysis could have provided additional insights into pediatric HIV infection in the CSF compartment.

Line 333-343 “Lastly, our analysis reflects the viral population diversity within the sample; however, Sanger sequencing has inherent limitations in detecting and interpreting quasispecies. The high mutation rate of HIV inevitably leads to sequence ambiguities, and despite manual chromatogram review, some errors may persist, particularly at low-quality positions. The limited depth of Sanger sequencing makes it challenging to distinguish low-frequency variants from sequencing artifacts. Although we retained ambiguous base calls where peak intensities were similar, this approach may not fully capture the viral complexity. For finer resolution of rare variants, next-generation sequencing would be more suitable. Nevertheless, for the neuropathogenic amino acid positions analyzed, no ambiguities were present, and we are therefore confident in the data presented for these specific signatures. ”

  1. No matched comparison between virus isolated from the CNS and PBMCs from the same patients. This is particularly important because of the point below.

Response: Due to the retrospective nature of our study, we had access only to cerebrospinal fluid (CSF) samples for each participant. Consequently, we were unable to perform direct comparisons between viral sequences from the CNS and PBMCs. We acknowledge that such comparisons are important, especially given the potential for HIV compartmentalization in the CNS. Previous studies have demonstrated that HIV variants in the CSF may differ from those in the blood, indicating local replication and evolution within the CNS. These differences can have implications for understanding the pathogenesis of HIV-associated neurological complications and for the development of targeted therapies. For that reason, we compared our sequence data to that of PBMC/blood derived sequences on the Los Alamos sequence datatbase to infer potential differences. While our study did not include matched PBMC samples, it provides valuable insights into the viral variants present in the CSF of pediatric patients with HIV. We hope that our findings will serve as a foundation for future studies that can include matched CNS and PBMC samples to further elucidate the dynamics of HIV compartmentalization and its impact on disease progression.

We also listed this as a limitation of our study.

Line 324-330 “We also compared our CSF-derived sequences to blood-derived sequences from other participants using data from the Los Alamos HIV Sequence Database. While this comparison provided some context, a direct comparison between our CSF-derived sequences and blood-derived sequences from the same patients would have been more informative. However, due to limited sample availability, this was not possible. Future studies should consider investigating matched CNS and PBMC samples to further elucidate the dynamics of HIV compartmentalization and its impact on disease progression.”

  1. It is not clearly defined that “X” in the protein coding sequences provided in the supplemental indicates a stop codon. Three of the four Tat open reading frames and all three of the Vpr open reading frames have stop codons, and several sequences have 2+ stop codons. This likely indicates that these sequences were isolated from replication defective virus (especially the stop codon at position 2 in Tat isolate SA_2). These observations underscore the need for more rigor in this study to assess the true virus diversity in this “cohort.”

Response: We appreciate the reviewer’s observation regarding the use of “X” in our supplemental protein sequences. We would like to clarify that in our dataset, “X” denotes amino acid positions with ambiguous base calls resulting from mixed nucleotide signals in the Sanger sequencing chromatograms. These ambiguities likely reflect underlying viral quasispecies diversity within the CSF samples, rather than the presence of stop codons or replication-defective viruses. We acknowledge that the use of “X” may have caused confusion, as standard conventions typically represent stop codons with an asterisk (*) or the label “Stop.” To avoid further misunderstanding, we have updated the supplemental materials to include a legend explicitly stating that “X” indicates ambiguous amino acid residues. Additionally, we reviewed the sequences submitted to GenBank and confirmed that no premature stop codons are present within the open reading frames. We also recognize that our use of bulk PCR and direct Sanger sequencing limits resolution of individual viral variants, which may contribute to perceived sequence ambiguity. This limitation is discussed in lines 333–343 of the manuscript. We agree that more rigorous methods such as clonal or deep sequencing should be employed in future studies to provide a clearer understanding of viral diversity in pediatric CSF. This comment is in line with comment 3.

  1. Given that 6/7 of the isolates here have stop codons, and thus are by default selected against, what can be gained from the phylogenetic comparisons with sequences in the LANL database given that those isolates do not have stop codons? In other words, polymorphisms in these deleterious proteins may not be functionally relevant as it could be a dead virus.

Response: We appreciate the reviewer’s observation regarding the interpretation of our phylogenetic comparisons. As previously clarified, the “X” residues in our Tat and Vpr protein sequences represent ambiguous amino acid positions due to mixed nucleotide signals in the Sanger sequencing chromatograms, not actual stop codons. These ambiguities likely reflect underlying viral quasispecies diversity within the cerebrospinal fluid (CSF) samples. Upon thorough review, we have confirmed that no premature stop codons are present in the open reading frames (ORFs) of our sequences.

In line with our earlier comments, we believe that the phylogenetic comparisons with sequences from the Los Alamos National Laboratory (LANL) HIV Sequence Database remain significant. These comparisons provide insights into the evolutionary relationships and potential functional implications of the viral variants present in the CSF of pediatric patients. Even in the absence of matched peripheral blood mononuclear cell (PBMC) samples, these analyses can highlight unique or conserved features of the CSF-derived sequences.

While our study did not employ clonal sequencing, the consensus sequences obtained through bulk PCR and direct Sanger sequencing still offer valuable information about the predominant viral populations in the CSF. Understanding these populations is crucial, especially given the role of the central nervous system as a reservoir for HIV and its implications for disease progression and treatment strategies. This comment is also addressed by comment 2 and 5.

Reviewer 2 Report

Comments and Suggestions for Authors

This work highlights an important difference in the viral pool sequences within the CNS and circulatory systems. This is of high significance for the general readership.

My average rating on the scientific soundness simply reflects the low numbers sampled, which the authors acknowledge. Nothing can be done about this aspect. but I am just noting that this is an area that can be improved with larger cohort size in future work.

Overall the manuscript is easy to read. One suggestion is to make some graphical representations of the data presented in the results sections as percentchanges in amino acids and different positions.

Some form of graphical summary of how and where the sequences differ in the various compartments compared to perhaps adult samples would strengthen the overall impact and help readability.

The figures of the phylogenic trees were blurry, and the fonts size can be increased for better visibility. Also can color be added to the phylogenic trees to highlight regions of similarity/differences that can be compared among and between the different figures?

Overall, the manuscript reads well, but graphical representation of the data in the results section, I believe would improve the readability and impact of the findings.

Author Response

13 May 2025

We would like to extend our sincere thanks to the reviewers and the editorial board for their careful evaluation of our manuscript. We are grateful for the constructive comments and valuable suggestions, which have helped us to improve the clarity, focus, and overall quality of the work. We appreciate the opportunity to revise our manuscript in response to the feedback, and we have carefully addressed each point below. The revisions made in response to their feedback are detailed below and have been appropriately highlighted throughout the manuscript.

Reviewer 2

This work highlights an important difference in the viral pool sequences within the CNS and circulatory systems. This is of high significance for the general readership.

My average rating on the scientific soundness simply reflects the low numbers sampled, which the authors acknowledge. Nothing can be done about this aspect. but I am just noting that this is an area that can be improved with larger cohort size in future work.

  1. Overall the manuscript is easy to read. One suggestion is to make some graphical representations of the data presented in the results sections as percent changes in amino acids and different positions.

Response: We appreciate your suggestion to enhance the manuscript with graphical representations illustrating percent changes in amino acids across different positions. In response, we have included sequence logos as supplementary figures (Supplementary Figures 2 and 4) to visually represent the frequency of specific amino acids at each position within the sequences analyzed. These sequence logos provide a clearer depiction of amino acid variability and conservation across the alignment. To maintain the concise format of the main manuscript, we have opted to present these visualizations in the supplementary materials. We believe this approach effectively addresses your recommendation while preserving the brevity of the primary report.

  1. Some form of graphical summary of how and where the sequences differ in the various compartments compared to perhaps adult samples would strengthen the overall impact and help readability. The alignments (figures 1 & 2) presented in the supplementary information provide a graphical representation of the differences between the CSF and blood compartments.

Response: We appreciate the reviewer’s suggestion to include a graphical summary comparing sequence differences across compartments and to adult samples. Our study specifically focuses on pediatric HIV-1 sequences, aiming to characterize viral diversity within the central nervous system (CNS) and peripheral blood mononuclear cells (PBMCs) in children. Given the unique virological and immunological context of pediatric HIV infection, and the limited availability of matched adult CNS-derived sequences, particularly for Tat and Vpr, direct comparisons to adult samples fall outside the scope of the current analysis. Furthermore, a search of the Los Alamos HIV sequence database revealed insufficient numbers of CSF-derived Tat and Vpr sequences from adult individuals, precluding a meaningful comparative analysis.

  1. The figures of the phylogenic trees were blurry, and the fonts size can be increased for better visibility. Also can color be added to the phylogenic trees to highlight regions of similarity/differences that can be compared among and between the different figures? Overall, the manuscript reads well, but graphical representation of the data in the results section, I believe would improve the readability and impact of the findings.

Response: We thank the reviewer for their helpful feedback regarding the clarity and presentation of the phylogenetic trees. Upon insertion into the manuscript template, the figure resolution was inadvertently reduced, affecting overall clarity. We apologize for this oversight. High-resolution versions of all phylogenetic figures will be included in the final submission to ensure optimal visibility and detail.

In response to the reviewer’s suggestions, we have updated the figures to increase font sizes for improved readability. Furthermore, we have introduced color coding to enhance interpretability: study-specific sequences are now highlighted in red, reference or comparison sequences in black, and branches with bootstrap support values greater than 70% are marked in blue. These visual enhancements are intended to aid in the identification of regions of similarity and divergence across the phylogenetic trees and to improve overall readability and impact of the results.

Round 2

Reviewer 1 Report

Comments and Suggestions for Authors

I stand by my initial assessment that this study does not contain the necessary rigor to make meaningful conclusions. The authors confirm that their bulk sequencing results are ambiguous at multiple positions and that the sample size is insufficient. Again, I understand that getting CSF samples from pediatric patients is challenging, but without additional data there is no way to eliminate sampling bias or make strong conclusions regarding the prevalence of the 4 Tat and 3 Vpr sequences reported here. 

Author Response

Response:

We thank the reviewer for their continued assessment and appreciate the acknowledgment of the challenges associated with obtaining pediatric CSF samples. While we recognize the limitations in sample size and the ambiguity at certain positions in the bulk sequencing data, we believe that these findings still offer novelty and stimulate important directions for future research.

The presence of specific Tat and Vpr sequences across our samples, despite the small cohort, suggests potential biological relevance that warrants reporting. Although we are cautious in our interpretation, we believe that highlighting these sequences, particularly in the context of such difficult-to-obtain samples, provides valuable preliminary insights and lays a foundation for future, larger-scale investigations.

Importantly, we have taken care throughout the manuscript to clearly frame our findings as exploratory and to avoid overstating our conclusions. Our intention is to contribute to the limited existing data on pediatric CSF-derived HIV sequences and to encourage further research in this critically understudied area.